# A Comprehensive Insight into the Role of Exosomes in Viral Infection: Dual Faces Bearing Different Functions

**DOI:** 10.3390/pharmaceutics13091405

**Published:** 2021-09-04

**Authors:** Mabroka H. Saad, Raied Badierah, Elrashdy M. Redwan, Esmail M. El-Fakharany

**Affiliations:** 1Protein Research Department, Genetic Engineering and Biotechnology Research Institute (GEBRI), The City of Scientific Research and Technological Applications (SRTA-City), New Borg EL Arab, Alexandria 21934, Egypt; mabrokasaad89@gmail.com (M.H.S.); rredwan@gmail.com (E.M.R.); 2Biological Science Department, Faculty of Science, King Abdulaziz University, Jeddah 21589, Saudi Arabia; rbadierah@kau.edu.sa; 3Medical Laboratory, King Abdulaziz University Hospital, King Abdulaziz University, Jeddah 21589, Saudi Arabia

**Keywords:** extracellular vesicles (EVs), pathogenesis and immune modulation, immune evasion

## Abstract

Extracellular vesicles (EVs) subtype, exosome is an extracellular nano-vesicle that sheds from cells’ surface and originates as intraluminal vesicles during endocytosis. Firstly, it was thought to be a way for the cell to get rid of unwanted materials as it loaded selectively with a variety of cellular molecules, including RNAs, proteins, and lipids. However, it has been found to play a crucial role in several biological processes such as immune modulation, cellular communication, and their role as vehicles to transport biologically active molecules. The latest discoveries have revealed that many viruses export their viral elements within cellular factors using exosomes. Hijacking the exosomal pathway by viruses influences downstream processes such as viral propagation and cellular immunity and modulates the cellular microenvironment. In this manuscript, we reviewed exosomes biogenesis and their role in the immune response to viral infection. In addition, we provided a summary of how some pathogenic viruses hijacked this normal physiological process. Viral components are harbored in exosomes and the role of these exosomes in viral infection is discussed. Understanding the nature of exosomes and their role in viral infections is fundamental for future development for them to be used as a vaccine or as a non-classical therapeutic strategy to control several viral infections.

## 1. Introduction

Extracellular vesicles (EVs) are a heterogeneous group of lipid-bound vesicles, which are derived from the plasma membrane or endosomes and secreted by almost all cell types into the extracellular lumen [1,2]. EVs are categorized into three main subtypes: exosomes (30 to 100 nm), microvesicles (~100–1000 nm), and apoptotic bodies (~500–4000 nm) based on their size, content, biogenesis, and function. These EVs subtypes have been detected in different biological fluids such as cerebrospinal fluid, saliva, blood, breast milk, ascetic fluid, amniotic fluid, seminal fluid, and urine [3,4]. At first, EVs are considered as cell debris, but recently they have emerged as important mediators in intercellular communication and they are involved in numerous physiological and pathological processes [5,6] such as inflammation and immune response [7], neuron-glia communication and myelination [8,9], infection [10,11], and cancer [12,13]. In addition, EVs play a crucial role in viral infection influencing viral entry, transmission, and immune evasion [14,15,16,17], as they serve as an important intercellular communication tool between uninfected and infected cells [15,18]. Viruses and EVs have common biogenesis pathways, so they are considered to be close relatives. Moreover, EVs shed from infected cells can either prompt an antiviral response or, on the contrary, increase viral infection [19].

In recent decades, exosome is one of the EVs that received great attention in different fields such as gene and drug delivery and prognostic and diagnostic biomarkers. Although the Greek word “exosome” comes from the “Exo” (“outside”) and “soma” (“body”), the accurate meaning is not understood immediately from the word’s parts and the term needs more explanation to be clear to non-specialists. The term “exosome” was first given to mobile DNA elements, then Mitchell and colleagues used this term to name an RNA processing body in the cell [20]. In the early 1980s, two teams of researchers discovered small vesicles formed via inward budding inside of the endosomal membrane while studying the reticulocytes culture media [21,22,23]; then at the end of the 1980s, Dr. Rose Johnstone gave the term “exosomes” to these small vesicles [24,25]. The International Society for Extracellular Vesicles (ISEV) recommended the term “extracellular vesicles” on the nomenclature of non-replicative, lipid bilayer naturally released vesicles from different cells as this term has a very clear meaning to non-specialists and specialists alike [26]. Regardless of the argument between scientists on the nomenclature and their bias to specific terms as we observed in different reviews in the literature, we will focus our discussion on the role of exosomes in viral infection.

## 2. Extracellular Vesicles Overview

Extracellular vesicles are considered one of the intercellular communication mechanisms derived from different cell types and they act as crosstalk between cells. These cell-derived membrane vesicles have a complex cargo containing proteins [27], lipids [28], and nucleic acids [29]. Different studies hypothesized that these cargoes are delivered to both local and distant cells, at which they exert their function. EVs are primarily classified into three main classes: exosomes (30 to 100 nm), microvesicles (~100–1000 nm), and apoptotic bodies (~500–4000 nm) and the differences between EVs are summarized as follows; apoptotic bodies are the largest extracellular vesicles (~500–4000 nm) that are observed during programmed cell death or apoptosis [30]. Apoptosis progresses through a series of stages which end with cell disintegration and enclosing cellular content in distinct membranous vesicles, named apoptosome or apoptotic bodies. In addition, the cargo of apoptotic bodies is characterized by the presence of different cellular organelles and/or nuclear content [31], [32]. Microvesicles (MV), also termed microparticles or ectosomes, emerged from the outward budding of the plasma membrane and have ~100–1000 nm in size [33]. MVs’ size, content, their formation from the plasma membrane, and their membrane-specific antigens, are the most distinguishing factor from apoptotic bodies [25]. Exosomes (our interesting point) are the smallest extracellular secretory nanovesicles with an estimated density between 1.13 and 1.19 g/mL and their size ranges from 30 to 100 nm [34]. Finally, the International Society of Extracellular Vesicles (ISEV) has recommended the generic term “EVs” for the vesicles derived from the cell due to the absence of specific markers and purification challenges for each EV subtype [26].

In recent decades, exosomes have acquired a large interest owing to their active role in the communication process between the cells [35]. They are derivatives of the cell’s endosomal process and form the multivesicular nanobody that tempers with the plasma membrane to excrete exosomes to the extracellular space [36]. Virtually, all cell types can release these nano-vesicles with varying levels, upon the fusion of the plasma membrane with multivesicular bodies [37,38,39]. Recently and contrary to what was initially thought, it is well established that exosomes are not the cell’s trash bags while they serve as imperative nano-vehicles for the transferring of specific viral cargo to inside and outside the host cells [40]. Based on their viral cargo, these exosomes can facilitate certain intercellular communication pathways [7,41]. Despite the mode of how this viral cargo is selected for packaging into these nano-vesicles intended for secretion, it remains unclear, and the endosomal membranes are supposed to play an essential role in this pathway [42,43]. After inward budding from early endosomal compartments, exosomes can form by packaging into the late endosomal membranes [22]. Then, both late endosome membranes and plasma membrane fuse leading to the release of the exosomes into the outside space of the cells.

It was established that exosomes exist in most secretions, including plasma, serum, saliva, tears, urine, semen, sweat, breast milk, and cerebrospinal fluid; besides, they are secreted in the supernatant of cell cultures [22], which is not limited to the vertebrates but is present in most invertebrates animals [44,45]. Depending on the status of the host cells, exosomes are composed of variable contents, and many studies established that exosomes can encapsulate multiple types of lipids and several kinds of proteins. These proteins include proteins that are complexed in vesicle formation, integral membrane proteins, membrane fusion-related proteins, proteins associated with cell metabolism and the cytoskeleton process, both classes I and II ingredients of the major histocompatibility complexes (MHC), and the cell surface proteins integrated with oncogenesis process [46,47]. Furthermore, most types of nucleic acids like DNA, long non-coding RNAs (lncRNAs), miRNA, and mRNA have been identified to be involved in the structure of exosomes [48,49]. Additionally, membranes of exosomes play an important role in the protection of their viral cargo from degradation by host enzymes; besides, they can provide other smart characteristics including high biocompatibility, overcome biological barriers, and low immunogenicity [50,51]. Exosomes provide precise intercellular communication by regulating the various levels of pathological and physiological processes by transmitting biological signals between the host cells. Besides the role of exosomes in pathogenesis, many specific components in exosomes can play critical roles in anti-viral activities via stimulating the antiviral immune responses or through inhibition of viral replication directly [52]. All these features of nano-vesicles encourage researchers and scientists to highlight the importance of exosomes.

## 3. Molecular Structure of Exosomes

Exosomes are multiform nano-vesicles with a diameter ranging from 30 to 100 nm which have a cup-shaped appearance on scanning electron microscopy. Exosomes classically consist of luminal cargo, including nucleic acids (DNA, RNA), lipid-derivatives, proteins, peptides, and amino acids enclosed in a lipid bilayer membrane (Figure 1). Exosomes act as transport vehicles and protective barriers to the luminal cargo from the tough extracellular surroundings [4]. The composition of this luminal cargo contains derivatives of cytosolic proteins from the donor host cell [53,54]. Advanced proteomic analysis techniques together with the high-resolution investigation by electron microscopy revealed the structure of exosomes released from various host cells [46,55]. The structure of the lipid bilayer derivatives of exosomes varies from the structure of the lipid of the plasma membrane of the host cell [56]. Exosomes are comprised of a rich level of lipids including sphingolipids, PS, cholesterol, and primarily ceramide [57]. Remarkably, membranes of exosomes do not involve lysophosphatidic acid [58], despite that lysophosphatidic acid has been identified in intraluminal vesicles and is supposed to be essential, together with Alix, for their formation [59]. Additionally, lipid membranes of exosomes also comprise subdomains of detergent-resistant lipid, i.e., subdomains enriched in sphingolipids and cholesterol (rafts). These lipid rafts are also enriched in numerous proteins such as flotillins that appear to bind to rafts [4]. Meanwhile, exosomes are containing protein derivatives including membrane fusion and transport proteins, multivesicular body formation proteins, tetraspanins, adhesion proteins, heat shock proteins, antigen presentation (MHC class molecules), and lipoproteins [54,60]. According to the type of infections, the composition of exosomal membranes is varied including modifications in the content of lipids and proteins and even spatial structure inversions. The quantity of exosomal proteins has been exposed to be changed under conditional stress or pathological status [61].

Furthermore, the outer surface of exosomes also contains glycan and polysaccharide derivatives, mainly include of mannose, polylactosamine, α-2,3- and α-2,6-sialic acids, and complex N-linked glycans [62,63]. Exosomes also contain nucleic acids including DNA and RNA such as miRNAs, some non-coding RNAs, and mRNAs [64]. Consequently, exosomes are categorized as bioactive nano-vesicles that contribute to intercellular communication by transferring their cargo to other host cells. By these facts, exosomes are involved in enormous pathways of biological processes, such as but not limited to those involved in the immunomodulatory process [65]. Furthermore, exosome cargo can contain some components of pathogenic microbes that are capable of enhancing immune and proinflammatory response [66]. This is because exosomes are formed intracellularly by the invagination process of the endosome membrane through the generation of an intraluminal vesicles bud into the multivesicular bodies in the cytosol. This invagination process seizes a large quantity of cytosol, involving the derivatives of enclosed proteins and RNA. Whereas exosomes comprise a collective amount of proteins regardless of host cell origin and topical investigations have revealed a cell kind-specific signature in exosomes [54].

## 4. Exosomes Biogenesis and Their Role in the Immune Response to Viral Infection

Exosomes generated by the endosomal route and this process initiated with inward budding of the plasma membrane resulted in early endosomes formation, followed by invagination of early endosomes membrane and enclosed the surrounding lumina with cytoplasmic content resulted in intraluminal vesicles (ILVs) formation [67]. Finally, the late endosome comprised dozens of ILVs which are termed as multivesicular bodies (MVBs). The fate of MVBs is the fusion with the plasma membrane that resulted in exosomes releasing, being transported to lysosomes for degradation of their cargo, or being delivered to the trans-Golgi system for endosome recycling [68]. The factors that control the fate or direction of MVB are not completely understood; however, previous studies reported that cholesterol-enriched MVBs undergo exosome releasing while the others undergo degradation. In addition, the two Rab family constituents, Rab27A and B, induce MVBs translocation to cell margin then sensitive factor attachment protein receptor (SNARE) complex facilities MVBs fusion with the cell membrane to release exosomes [69,70]. Endosomal-sorting complex required for transport (ESCRT) plays a crucial role in exosome biogenesis and releasing process [71]. ESCRT system involves four complexes known as ESCRT-0, ESCRT-I, ESCRT-II, and ESCRT-III with associated proteins (Tsg101, ALIX, and VPS4). During the biogenesis process, each complex has the role as follows: ESCRT-0 is recruited by ubiquitinated cargo to the lipid domain and initiates the pathway, ESCRT-I and ESCRT-II complexes trigger the deformation of membrane resulting in buds or stable membrane neck and this is also responsible for the recruitment of Vps4 complex to ESCRT-III which separates or scissors from the cytoplasmic membrane [72]. In addition, several studies discussed exosome biogenesis and their cargo loading in the route of ESCRT-independent pathway, which comprises lipids and associated protein as tetraspanin [73]. While proteins required ESCRT complexes to be loaded into exosomes, RNA sorting through a process depending on self-organizing lipid and cargo domains as a specific RNA sequence has an affinity for the phospholipid bilayer, which is influenced by hydrophobic modifications, lipid rafts, and sphingosine concentration in membrane rafts [74].

These released nano-vesicles may enhance immune response and present antigens of viral pathogens through a cellular immune response. Meckes and Raab-Traub [15] revealed that exosomes have numerous features in common with enveloped viruses such as biogenesis, biophysical characteristics, and sorting in cells. Recent studies defined the nano-vesicle-mediated intercellular transfer of functional cellular proteins; mRNAs and miRNAs have exposed further similarities between viruses and cellular nano-vesicles. They also showed that the editing enzyme of apolipoprotein B mRNA catalytic subunit 3G, -a cytidine deaminase that contributes to the antiviral cellular response against retroviruses, might be preventing HIV-1 replication through an accumulation of exosomes in neighboring host cells. Izquierdo-Users et al. [75] revealed that HIV-1 sorts all particles and antigens in exosome-like vesicles after fusing with DCs uses. They revealed also that HIV-1 uses a cluster of DCs as a transit location in the non-replicative phase. Van Dongen et al. [76] showed that exosomes provoke viral infection through bearing viral antigens and transferring their cargos to CD4 + T cells (Table 1).

Furthermore, some viruses might also operate with the machinery of vesicular trafficking for their assembly, egress, and uptaking [17,121]. Recent studies revealed that HIV-1 has been shown to exploit the lipid raft domains, endosomal sorting complex required for transport-II (ESCRT), and Rab GTPases constituents, all of which are included in exosome biogenesis [122,123]. Exosomal tetraspanins, particularly CD81 and CD63, interact with egress HIV-1 Gag to assist in virion. Mori et al. [124] revealed that Human herpesvirus 6 (HHV-6) virions have been established to use the same pathway of induction for the formation of multivesicular bodies and egress happens by an exosomal release pathway. Intraluminal vesicles transfer plasma membrane invaginations to the endosomal set via clathrin-independent and clathrin-mediated endocytosis process (back fusion process) making exosomes capable of harboring both extracellular and intracellular ingredients [80,125]. Exosomes are characterized by their ability to provide snapshots of the microenvironment, making them serve as an effective resource of biomarkers. This exosome feature is owing to the great cross-over between viral entry and normal macromolecule entry, exosome biogenesis, release, and cycling within the host cell [80,125]. Furthermore, there are other enriched ingredients in exosomes such as lipid raft-associated proteins (proteins included in membrane trafficking including the tetraspanins CD63, CD9, and CD81, transferrin, and caveolins) which have been shown to bind to ESCRT machinery [15,126,127]. One method for viral intonation of exosome secretion is by directly binding with the machinery included in exosome biogenesis (ESCRT-dependent method). However, ESCRT-independent processes can also occur by which lipids, proteins, and nucleic acids can originate from the endosomal pathway mechanism. For instance, oligodendrocytes initiate exosome formation through the ceramide machinery pathway [128], whereas other host cell kinds depend on the oligomerization of tetraspanin complexes [129,130]. Additionally, although the reduction of some ESCRT ingredients might revoke exosome formation, it does not entirely knock it out [131,132]. Specific candidates of the Rab GTPases family, a family of highly conserved proteins that regulate many processes of membrane fusion and vesicular trafficking in eukaryotes, are also concluded in exosome production as indicated by their extraordinary richness in extracted exosomes [11,133]. Recently, it has been shown that activation of Rab GTPases and interfering with their levels enhances the secretion of exosomes from host cells. Depending on the kind of host cells, Rab5, Rab7, Rab11, Rab27, and Rab35 were all associated with the secretion process. It was found that overexpression of Rab5 inhibits the development of endocytosed material from early endosomes, leading to markers reduction of exosomal releases such as CD63, Alix, and syndecan, and this negative impact was liberated by the Rab7 overexpression [134]. Being GTPases, each Rab stimulation is dependent on the calcium influx process such as in the event of Rab 11 in the K562 cells, which might include SNARE proteins [135,136,137]. Overall, there are many players inside the host cell that donate to the endosomal process and, eventually, to the secretion of exosomes, promoting emphasis on the significance of this process in standard biology. In this way, viral pathogens also have other advanced machinery modules to involve their viral cargo at every step of exosome biogenesis [17].

## 5. Exosomes Purification and Characterization

### 5.1. Ultracentrifugation Techniques

Ultracentrifugation (UC) is considered the gold standard isolation technique for exosomes and the most widely applied isolation method for exosome extraction and separation. Separation of desired components using ultracentrifugation technique is mainly based on the difference in size and density of each component in mixture solution, so large-dose sample constituents which significantly differ in sedimentation coefficient can be effectively separated using this method [138]. Johnstone et al. [139] first used the ultracentrifugation technique to separate exosomes from reticulocytes and Thery et al. [140] optimized this method. The ultracentrifugation technique involves two main steps: (1) getting rid of large-size extracellular vesicles, cell debris, and dead cells through a sequence of continuous low-medium speed centrifugation, followed by separating exosomes at higher speed centrifugation (100,000× *g*). In addition, studies have reported that the purity and yield of target exosomes are affected by centrifugal parameters including centrifugation force, time, and rotor type [138,141]. This method does not need to label exosomes, which can avoid cross-contamination, but it is not conducive to downstream analysis due to its time consumption, high cost, structural damage, aggregation into blocks, and lipoprotein co-separation [142]. Furthermore, the density gradient centrifugation is usually applied within ultracentrifugation to enhance the purity of the exosomes. The density gradient centrifugation is divided into two main classes: one of them using sucrose as a gradient medium that is widely applied in research. However, this method is not successful in the separation of retroviruses and exosomes which nearly have the same density and size. For this purpose, Cantin et al. [143] reported that the iodixanol gradient is effective in the exosomes separation from HIV-1- infected cells with high purity yield. Although, the exosome purity is the advantage of the density gradient centrifugation; decreasing the sedimentation rate of exosomes due to the high viscosity of sucrose is a point against this technique [144].

### 5.2. Polymer Precipitation

This technique was first used to separate viruses [145]. As exosomes are virus-sized membranous particles and share viruses in their biophysical characteristics, this technique is often used in scientific research to separate and purify exosomes. This method usually uses polyethylene glycol (PEG) to reduce the exosomes solubility and precipitate them followed by centrifugation. Rider et al. [145] recommended the modified polymer co-precipitation (ExtraPEG) method for exosome separation and reported that this technique is cost-effective and outperforms the ultracentrifugation technique in terms of recovery and purity depending on comparative study between three technologies of commercially available commercial kits, ultracentrifugation, and modified polymer co-precipitation (ExtraPEG). This method has the advantage of processing large doses of samples and is relatively easy to operate with a short analysis time. Although, false positives may be obtained within this method and eradicating the produced polymer is hard.

### 5.3. Size-Based Isolation Techniques

Size exclusion chromatography (SEC) is a technique applied in exosomes separation from other components in biological samples depending on their size difference as the macromolecules cannot pass in gel pores and subsequently eluted, whereas the small molecules trapped in the gel pores which finally eluted with the mobile phase. The advantages of this technique are easy, cheaper, fast and the isolated exosomes are uniform in size, and their biological features are not significantly damaged. The disadvantage of this method is low purity as other components have a similar size [142]. Recently, Exo-spin exosome purification columns, EV Second purification columns, and EV separation columns are commercially available. In addition, the ultrafiltration technique using membranes with diverse molecular weight cutoffs is used for exosomes separation and also faced the problem of low purity [146].

### 5.4. Immunoaffinity Chromatography (IAC)

Immunoaffinity chromatography is another technique used for exosomes separation and purification depending on the affinity or binding of ligands and antibodies to separate them from a biological fluid. The efficiency of this method is mainly attributed to the affinity of desirable components and ligands, matrix carriers, and elution conditions. In addition, the targeted biomarkers used for exosomes separation and purification by this method should be high-abundance on their surface, for example, ESCRT complex-related protein and four-transmembrane protein superfamily. The advantages of the IAC technique are high purity, high sensitivity, strong specificity, use for quantitative and qualitative determination of exosomes, and the starting sample volume not being limited when magnetic beads are used [147,148]. In addition, the quantitative analysis showed that the exosomes yield obtained from this method is equal to that of ultracentrifugation, but the applied sample is much smaller as the RNA amount extracted from 400 μL plasma using IAC is equaled the amount obtained from 2.5 mL sample using ultracentrifugation technique [149]. In contrast, this method faced different problems that limited its propagation as exosomes extracted by the IAC technique required harsh storage conditions, were contaminated with interfering proteins, and were not an effective technique for large-scale purification. Recently, commercially available kites have been developed based on the above-discussed method and scientists exerted their efforts to overcome the problems faced by exosomes purification, for example, development of a new micro-vortex chip technique, that is integrated with modified nanoprobe Morpho Menelaus butterfly wings [148]. Although several techniques have been developed for exosomes purification, they cannot meet all requirements and the combination between these techniques is the best. The exosome characterization process required developed and optimized guidelines to determine whether the extracted elements are exosomes. International Society for Extracellular Vesicles (ISEV) recommended [26] two main principles for the identification of extracted exosomes, the first point is the determination of two types of proteins (like cytosolic proteins recovered in EVs and transmembrane or GPI-anchored proteins associated to plasma membrane and/ or endosomes) and the second point is the assessment for exosomes purity and they are free from associated non-EV structural protein components during extraction from biological fluids. Exosome characterization approaches are generally divided into two types: inclusion characterization (such as lipid raft and membrane protein) and external characterization (morphology and particle size determination). Several investigations depended on three levels for characterization of isolated exosomes, including determination of exosome morphology using a Transmission electron microscope (TEM), determination of exosome surface protein markers through Western Blot analysis, and identification of exosome size through Nanoparticle Tracking Analysis Technology (NTA) [150,151].

## 6. Exosomal Pathway and Viral Pathogenesis

There are over one hundred million people worldwide who are affected by microbial (bacteria, fungi, viruses, parasites) diseases such as tuberculosis, acquired immune deficiency syndrome, and malaria [152]. Throughout recent decades, information about exosomes has progressed in a diversity of scientific directions, predominantly regarding viral infections. Exosomes were found to be either inhibited or to promote the process of viral infection and pathogenicity depending on the pathogen and its target. In all circumstances, exosomes can make possible communications between pathogens and host cells or between host cells [153]. Recently, several studies have recognized exosomes and their crucial role in viral pathogenesis and immunity [121]. Exosomes transfer antiviral elements and activating antiviral mechanisms in a variety of cells allowing the host to mount effective immune responses against different viruses [123]. Exosomes enclosing viral genomes can transfer them to susceptible cells aiding viral spread and evading immunity infiltration [76]. In certain circumstances, exosomes that comprise viral proteins or nucleic acids stimulate immune responses in myeloid cells [114]. Several studies on Human Immunodeficiency Virus type-1 (HIV-1), human T-cell lymphotropic virus (HTLV), Dengue Virus, and Hepatitis C Virus (HCV) have shown that exosomes shed from virus-infected cells are loaded with various regulatory factors, including cellular and viral proteins, miRNA assisting in regulating cellular responses, and producing infections in neighboring cells [121].

### 6.1. Viruses Hijack the Exosomal Machinery System (ESCRT and Rab GTPases)

Viruses are intracellular obligate parasites exploiting cellular replicating machines to produce new progeny. Recent studies have shown that some viruses can hijack vesicular trafficking machinery for their transmission, assembly, and egress [17,121]. HIV-1 and HIV-2 exhibit similar structural and molecular features with exosomes. They are enclosed by a lipid bilayer containing RNA species [154], proteins [22,155], lipids [156], and carbohydrates [157]. In addition, the density and size of both range from 1.13 to 1.21 g/mL [79] and 50 to 150 nm in diameter [61], respectively. Depending on these similarities, different studies have reported identical pathways of HIV-1 production and exosome biogenesis [39,154]. In approval of the Trojan exosome hypothesis, HIV-1 exploits the host ESCRT complexes for viral budding [15,123]. The association between HIV-1 Gag protein and tetraspanin suggests that HIV-1 assembly may involve lipid raft domains rich in tetraspanins [158]. In addition, HIV-1 Gag has been shown to interact with CD63 and CD81 located on the surface of some exosomes aiding in virion egress [78]. Human herpesvirus 6 (HHV-6) infections intensely increase MVB formation for virions survival and also egress together with exosomes exploiting the same pathway [124]. Moreover, HHV-6 glycoprotein gB was found in association with CD63, impressing the importance of the endosomal pathway for HHV-6 infection and assembly [124], but the account of this association for virus egress has not been proven. Over hijacking the ESCRT pathway, certain viruses can also exploit the Rab GTPase complexes for viral egress and complete their replication processes. Many negative-strand RNA viruses, like respiratory syncytial virus (RSV), Hantavirus, and influenza A virus (IAV), all have been exploiting the Rab pathway for transport through the plasma membrane [159,160,161,162]. In addition, interfering with Rab11 protein levels can promote or inhibit the release of exosome-carrying different compounds: anthrax toxin, transferrin, flotillin, and HSP-70 [159,163,164]. Hantavirus and influenza A virus (IAV) also appear to recruit the Rab11 pathway to their profit [160,162] while its depletion leads to a tenfold reduction in hantavirus production [161]. Another member of the Rab GTPase family, Rab27a, has been reported to be necessary for exosome biogenesis, mainly during the MVBs fusion with plasma membrane for exosome releasing [165,166]. The levels of Rab27a have increased with cytomegalovirus (CMV) infection and colocalize at assembly sites with the viral envelope components [167], but the changes and molecular mechanisms of the exosome production are not completely understood. In addition, HIV-1 proteins are found to interact with Rab27a, increasing exosome formation levels [168,169]. Herpes simplex virus 1 (HSV-1) appears to recruit Rab27a on intracellular transport and exocytosis steps as downregulation or depletion of Rab27a results in the reduction of HSV-1 production [170,171]. Although the concept of interaction between cells and stimuli to regulate the distribution, secretion, degradation, recycling, and the level of intracellular proteins is widely believed, Rab GTPase regulatory functions are still not completely understood [172].

### 6.2. Signatures of Different Viruses in Exosomes and Their Role as Vehicles for Viral Elements

The common features between the cellular endosomal/exosomal pathway and different viruses’ life cycles besides exploiting the exosomal pathway for virus profit encouraged several studies to identify viral signatures in exosomes not only in terms of viral diagnosis but also for demonstrating the mechanisms of viral pathogenicity. We now have a growing list of viral-specific components that have been identified in exosomes (Table 1). Moreover, functional analysis of excreted exosomes carrying viral components is beginning to shed light on how some viruses can modulate cellular processes as diverse as immune evasion, apoptosis, proliferation, and even viral infectivity (Figure 2). Among viruses family, human herpesviruses have received extensive studies, especially Kaposi’s sarcoma virus (KSV) and Epstein-Barr virus (EBV) that are oncogenic and associated with the development of several human malignancies [173]. Commonly, these viruses have been reported to recruit the exosome pathway to excrete several components ranging from different RNA species, including microRNAs (miRNA), messenger RNAs (mRNA), and small non-protein-coding RNAs to proteins [120,174,175,176]. In the case of EBV-infected cells, the released exosomes comprised an intense amount of viral miRNAs which identified for the first time, seem to be the smaller products of larger BamH1 EBV transcripts and work in association with cellular miRNAs to modulate target genes expression in recipient host cells [175,177,178,179,180]. Moreover, exosomes not only constitute as carriers of virus RNA species but also for viral-specific proteins as the exosomes shed from EBV- infected cells contain non-protein-coding EBV small RNAs (EBER-1/2) [91], viral envelop glycoprotein 350, and the latent membrane protein 1 and 2A (LMP-1, LMP-2A) [88,89,90,181]. The highly abundant EBV RNA polymerase II/III transcripts, EBER1/2, are expressed during the latency stage in all EBV infected cells. Although, the reason for their abundance in infected cells, or their excretion within exosomes, is still intriguing; one study indicated that they could activate Toll-like receptor 3 enhancing innate immune responses [182]. HIV-1, which belongs to the retroviruses family and is responsible for AIDS syndrome, is another virus receiving great attention. Retroviruses are slightly larger than exosomes; therefore, mature infectious HIV-1 particles cannot secrete within exosomes but are released together in the same fraction suggesting HIV-1 egress is partially mediated by the endosomal pathway [183]. Furthermore, exosomes derived from HIV-1-infected cells have been shown to incorporate different HIV-1 RNA species, including vmiR88, vmiR99, vmiR-TAR transcripts [80], and two HIV-1 proteins; Gag [184] and Nef proteins [185,186]. It has been reported that, among viral microRNA species, vmiR99 and vmiR88 stimulate macrophages signals (Toll-like receptor-8 (TLR8)), rising TNFα level [187] while vmiR-TAR transcripts have been suggested to augment HIV-1 replication through downregulation of apoptotic pathway in the recipient host cells [188]. HIV-1 Nef protein is interacted with exosomes upon anchoring into lipid rafts through its N-terminal myristoylation and basic amino acid residues of its alpha-helix-1 [189]. Lately, several studies have shown the crucial role of Nef protein in viral replication and infection through the transformation of resting bystander CD4+ T cells to HIV-1 susceptible cells [111,186,190] while HIV-1 Gag protein plays a role in HIV-1 assembly and egress through the interaction with the exosomal membrane tetraspanins, CD81 and CD63 [110,158,191,192]. In addition, exosomes incorporating viral co-receptors, CXCR4, and CCR5 may aid in HIV-1 spread and infection by secreting to other cells [193,194]. In the same way, human T-lymphotropic virus 1 (HTLV-1) appears to recruit the exosomal system as a vehicle to transport its active components including viral mRNA transcripts and functional proteins. A recent study revealed that the HBZ, Env, and Tax gene mRNA transcripts and trans-activator protein, Tax, released within the exosomes shed from HTLV-1-infected cells [118,195]. In the case of human T-lymphotropic virus 1 (HTLV-1)-associated myelopathy/tropical spastic paraparesis (HAM/TSP), analysis of the exosomes isolated from patient cerebrospinal fluid showed the presence of transactivator protein Tax, proposing that HTLV-1 may release specific cargo for modulating its microenvironment [196].

Recently several studies have supported the idea that exosomes act as an alternative tool for HCV transmission and infection. HCV-infected hepatocytes excreted exosomes within the complete HCV genome and able to transfer HCV subgenomic replicons RNA to recipient cells resulting in viral replication and infection [52,92]. Exosomes can successfully deliver HCV RNA to dendritic cells to establish viral infection [94] and stimulate IFN-α production [165]. Furthermore, Bukong et al. [93] reported that exosomes shed from HCV-infected patients have been comprised of negative-sense HCV RNA complex with miR-122, Ago2, and HSP90 [93]. Similarly, the human Pegivirus that was earlier known as the Hepatitis G virus recruits exosomes as a means to transport viral RNA to peripheral blood mononuclear cells and establish a productive infection [197]. In addition, exosomes derived from Ebola virus (EBOV) infected cells enhanced EBOV infectivity. These exosomes were loaded with cytokines contributing to EBOV pathology and EBOV matrix protein, VP40 which act on recipient cells leading to decrease monocytes and t cells viability [141,142]. As the number of exosomal components that is originated from cells and microbes is continually increasing, several online databases are launched to catalog exosome contents. Furthermore, recent studies are beginning to explore the cargo of exosomes derived from virally infected cells and their role in the pathogenesis of viral infections. The mechanism through which cellular and microbial components are selected to be packaged into exosomes and the possibility of using exosomes as biomarkers for many disease progression and different viral infections is the main challenge to be understood [198].

### 6.3. Role of Exosomes as Potential Mediators in Viral Pathogenesis

Exosomes derived from viral-infected cells export viral components together within the cellular one [80,198]. Besides, viruses excrete their elements in exosomes; they somehow control which cellular products are transported within the exosomes as the cellular component packed within the exosomes is different within the same non-infected cells [91,120,176]. Consequently, the viral components of the exosomal cargo are responsible for any pathophysiological effect on recipient cells. Although several data display that exosomes shed from viral-infected cells induce different processes including, cytokine modulation, immune evasion, transcellular spread, apoptosis, and proliferation, the exact molecular mechanisms through which these processes occur are not understood [11,34]. In the case of EBV, exosomes shed from EBV-infected B cells attack epithelial cells through caveolar-dependent endocytosis and their cargo induce physiological changes [199]. It has been reported that exosomes from EBV-immortalized lymphoblastoid cell lines (LCLs) and nasopharyngeal carcinoma (NPC) promote apoptosis or inhibit EBV-reactive CD4+ cells proliferation [90,200,201]; also, similar observations were reported with exosomes from NPC-xenografted mice and EBV-associated from nasopharyngeal carcinoma cases [202]. The cargo of these exosomes comprises viral and cellular elements as viral miRNAs, latent membrane protein-1 (LMP-1), LMP-2A, and cellular galectin 9 [18,89,202]. The EBV-LMP-1 is an oncoprotein that has a role in the immortalization of EBV-infected B lymphocytes [203]. LMP-1 of EBV-infected cells is functionally homolog of TNF receptors as it activates nuclear factor κB (the main transcription factor) through transmitting growth signals from cell membrane to the nucleus in TNF-receptor associated factor pathway [204,205]. In addition, the other EBV latent membrane protein, LMP-2A [206], plays a crucial role in EBV pathogenesis and latency or mediates EBV-infected B cells immortalization as it mimics a constitutively active B-cell receptor (BCR) activator even in the absence of BCR signaling [207,208]. Despite the secretion of EBV-latent membrane proteins within exosomes enhancing EBV pathogenesis in the recipient cells, the function of these proteins in the recipient cells and how they are selected to be excreted within exosomes is still one of the major challenges to be understood [90,200]. Lately, it has been reported that exosomes derived from EBV-infected B cells export Fas-ligand (Fas-L) to recipient cells where they induce apoptosis in a dose-dependent manner via an extrinsic pathway, and anti-Fas-L antibodies could block this process [87]. In addition, another study showed that Fas-L and MHCII molecules excreted within the exosomes derived from lymphoblastoid cell lines, prompting apoptosis in CD4+ T cells [201]. From these studies, the shedding of exosomes within signals which promote apoptosis and/or inhibit anti–EBV-infiltrating lymphocytes is a strategy through which EBV could evade the immune system.

Some viruses can exploit a mechanism of downregulating their lytic gene expression to escape the immune responses and establish latency in the infected cells [209] as the immune system is not triggered unless viral antigens are expressed [210]. Herpes simplex virus type 1 (HSV-1) is one of the *Alphaherpesvirinae* family primary replicating in mucosal epithelial cells and established their latency in sensory ganglia [211]. In the latency, abundant vmiRNAs have been identified without viral protein expression, and some of these vmiRNAs have a significant role in latency establishment through the suppressing of viral gene expression [212,213,214]. Recent studies have revealed that HSV-1 can spread its infection through exporting significant components within exosomes to the recipient cells such as vmiRNAs important for suppression of viral gene [215] while HSV-1 can suppress cell-cell transmission through excretion of antiviral factor, STING within exosomes to recipient cells in unfavorable conditions [83]. These strategies enable HSV-1 to persist and escape from immune eradication. In addition, L-particles are subviral non-infectious particles (not viral DNA or viral capsid) released from HSV-1-infected cells within exosomes to recipient cells [216,217] suggesting their role in facilitating viral infection and immunity infiltration via microenvironment modulation [218]. Similarly, some viruses transport molecules that neutralize antiviral and cellular host inflammatory factors within exosomes modulating their microenvironment [42,219]. In hepatitis B virus (HBV) infection, the serum was found to comprise subviral non-infectious elements at the level of 1000 times higher than mature infectious virus, which raises the question about the selectivity of HBV to secrete such non-infectious particles [113,220]. One of the most acceptable suggestions is that these non-infection particles act as a trap and switch the immune response to recognize infectious virions [14,113]. Furthermore, some members of the *herpesviridae* family, including herpes simplex virus type-1, human herpesvirus 6 (HHV-6), and cytomegalovirus (CMV) can recruit the endocytic pathway to complete their assembly and egress steps during their replication [80]. In HSV-1 infection, the interaction of enveloped glycoproteins (gD and gH) and tegument protein with Rab27a resulted in depletion of Rab27a and thus reducing viral egress and production and signifying Rab27a role in HSV-1 production [171].

The role of exosomes in HIV-1 infection is not completely known; several studies reported that exosomes can either inhibit or enhance HIV-1 infectivity and pathogenicity. Recent studies have shown the significant role of exosomes in HIV-1 infection. HIV-1 coreceptors CCR5 and CXCR4 have been reported to be transferred within exosomes from HIV-infected cells to uninfected, non-permissive cells, making them more susceptible to viral infection [193,194]. Recently, it has been reported that exosomes excreted from monocyte-derived macrophages enclosed a portion of HIV-1 virions and the infectivity of these enclosed virions toward CD4+ target cells is better than the HIV-1 virus particles [221]. The plasma cytokines levels are relatively increased in HIV-1 infection and several cytokines have been reported to be excreted within exosomes, playing a significant role in viral propagation and inflammation. The infection of peripheral blood mononuclear cells with exosomes purified from HIV-1 positive individuals resulted in stimulating CD38 expression on central memory CD8+ and CD4+T cells, which activated bystander cells and facilitated viral transmission and inflammation [222]. Recently it has been reported that exosomes excreted from HIV-infected cells containing molecules of MHC Class II, CD86, and CD45 may help in silencing immune response, thus supporting HIV replication [223], while exosomes released from HIV-1 infected CD8+ T cells inhibited transcription of CXCR4 and CCR5-tropic HIV-1 strains in the chronic and acute infection models [224]. As well, HIV-1 accessory protein, Nef is also excreted within exosomes and has a significant role in HIV-1 infection via assisting in CD4+ T cell depletion that is the target infection of HIV-1 [185]. Nef interacts with the cytosolic tail (CT) of MHC-I and CD4 resulting in the interruption of the intracellular trafficking process and making these proteins a target to multivesicular bodies and finally degradation by lysosomes. HIV-1 Nef exported within the exosomes modulates exosomal miRNA composition [186] and prompts the activation of quiescent CD4+ T lymphocytes to be permissive to HIV-1 infection and transmission [190]. Recently, in vitro study reported that exosomes shed from CD4+ T cells not from CD4- T cells well inhibit HIV-1 infection, proposing that exosomal CD4 interacts with HIV-1 envelope proteins, neutralizing HIV-1 and preventing its interaction with target cells [186]. This study also reported that this neutralization activity can be reversed through depleting CD4 exosomes excreting from Nef expressed CD4+ T cells [186]. Moreover, cellular defense enzyme, human cytidine deaminase APOBEC3G (A3G) has been reported to be exported within exosomes and gained recipient cell resistance against defective and wild-type HIV-1 infection [225]. As mentioned above, HTLV-1 recruits exosomes for transporting functional viral elements (HTLV-1 mRNA transcripts of HBZ, Tax, and Env proteins, pro-inflammatory mediators, and HTLV-1 Tax protein) together with cellular host protein (major histocompatibility complex class I A and E). The exposure of myeloid dendritic cells to exosomes enclosed Tax protein and derived from C81 cells resulted in significant elevation of cytokines level, including IL-6, IL-5, and IL-2 while, cell-free Tax stimulates the cytokines secretion of G-CSF, IFN, IL-17A, IL-12, and IL-10 from dendritic cells suggesting the exosomal trafficking role in HTLV-1 infection [118]. 

In HCV infection, exosomes not only act as a vehicle for HCV transmission to naive cells but also a place where virions escape from antibody neutralization. A recent study using transmission electron microscopy showed that HCV was existing in exosome-associated and exosome-free forms and exporting within exosomes has the advantage of protection against HCV antibody-mediated neutralization, making the transmission within exosome a significant mechanism by which HCV can escape from the immune system [92], [226]. Similarly, Hepatitis A virus (HAV) was found to be enclosed within membranous vesicles derivative from endosomal membranes and still infectious. These encapsulated HAV particles used the ESCRT pathway for their biogenesis which confers a protective mechanism against different neutralizing antibodies [172]. The level of an antiviral protein called interferon-inducible transmembrane proteins 3 (IFITM3) in the host cell is reversely related to the host susceptibility to infection with dengue virus serotype -2 (DENV-2). Recently, it has been reported that exosomes excreted from HepG2 or HUVEC cell lines transport IFITM3 protein to bystander cells. In the recipient host cell, these IFITM3-containing exosomes effectively inhibited DENV-2 infection in a dose-dependent manner. This study reported a significant role for IFITM3-enclosing exosomes in DENV-2 infection through the reduction of DENV-2 penetration into host cells without any effects on the binding or post-entry steps [95].

Many viruses are reported to hijack host machinery of exosome biogenesis to regulate their production and virus secretion and used Trojan horse stratagem to enclose new viral progeny within exosomes helping viruses to spread and harbor from host defense [227]. SARS-CoV-2 is the newest positive-strand RNA virus that employs the cell membranous machinery to, through its nsp1-10, create a double-membrane vesicle (DMV) just a few seconds after it is attached to the ACE2 receptor, where the virus replicates, assembles, and finally exports into extracellular [228]. The main ideas that raise the question about the involvement of exosomes in SARS-CoV-2 infection are (1) pathophysiologies of SARS-CoV-2 infection, mainly hyper activate immune response which promotes sepsis-like disease characterized by decreasing lymphocytic count and cytokine storm. (2) The exploitation of the TGN pathway (trans-Golgi network, which is a major secretory pathway sorting station in the endolysosomal pathway) through SARS-CoV-2 replication. (3) Recent studies reported that the pathogenesis of COVID-19 complications involved lipid metabolism as cholesterol metabolism [229,230]. Protein interactome analysis for SARS-CoV-2 displayed the interaction with Rab proteins (a part of the ESCRT pathway involved in exosome biogenesis) and strengthened the idea of involvement exosome in SARS-CoV-2 pathogenesis as many viruses interact with Rab proteins and hijack exosomes for pathogenesis [167,171]. Furthermore, high-throughput lipidomics of sera from infected people indicated that the lipid profile of derived exosomes enriched with gangliosides (GM3) and sphingomyelins and was deficient in Di-acyl glycerols (DAG). Gangliosides-enriched exosomes were likely the cause of lymphopenia as immune cells have a preference for such exosomes and these exosomes are strongly related to the disease severity [231,232]. Although Coronavirus disease 2019 (COVID-19), which was caused by the severe acute respiratory syndrome coronavirus 2 (SARS-CoV-2), is a newly born disease, it has reached a pandemic level with little information about its molecular targets, immunogenicity, and the role of epigenetics mechanisms as miRs and other noncoding RNAs in pathogenesis. Extensive studies suggest that these non-coding RNAs are packed within exosomes which shuttle them between different cell types, promoting sepsis and tissue injury. Recently, in vitro study helped us to predict the functional role of exosomes in the SARS-CoV-2 pathogenesis as it reported that transduction of SARS-CoV-2′s structural and nonstructural genes with lung epithelial A549 cells results in the discharge of exosomes enriched with viral RNAs; taking the human-induced pluripotent stem cell-derived cardiomyocytes (hiPSC-CMs) resulted in elevated inflammatory markers [233]. This result may interpret the possible mechanism of cardiac complications associated with SARS-COVID-19 infection that has mystified scientists. The SARS-CoV-2 infection resulted in the extensive inhibition and activation of protein kinases [234]; also, exosomes derived from virus-infected cells may be loaded with proteins that can stimulate the inflammatory system and cause different tissue injuries. One of the studies concerned with the development of a culture system permissive for SARS-CoV reported that human alveolar type II was susceptible to SARS-CoV infection and differentiation and SARS-CoV antigens were identified in the cytoplasm of infected cells. In addition, this study reported that electron micrograph detected virions in virus-induced secretory vesicles where the viral RNA increased with time indicating that SARS-CoV exploits it for their replication. The virions bud from the membranes of the ER–Golgi-intermediate is enclosed with secretory vesicles and released from cells by exocytosis [235]. As recently reviewed, the post-mortem histopathological analysis of samples taken from several organs of COVID-19-infected people revealed that the presence of SARS-CoV-2 particle and/or its proteome within the double-membrane vesicles (DMVs) of the host cells [228,236]. The electron microscopic examination of the renal tubular epithelium revealed the arrangement of coronavirus particles with their distinctive spikes in clusters [237]. Farkash et al. (2020) reported that an electron microscope examination for an autopsy renal sample from a COVID-19-infected patient showed an arrangement of viral particles in arrays indicating their intracellular manufactured and assembly. Furthermore, the presence of DMVs near the rough endoplasmic reticulum during viral assembly suggests that the assembly mechanism of the SARS-CoV- 2 viral is similar to that of SARS-CoV [238]. The outcome of these results recommended that the possible participation role of exosomal cellular transport in SARS-CoV-2 viral dissemination and serve as a valuable tool for the COVID-19 reactivation. However, the main infected organ of SARS-CoV-2 infection appears to be the lung, the detection of virion particles in urine [239], and stool [240] indicating their spread to other organs such as the kidney and intestine. Although SARS-CoV-2 is viremic, infection has been established in the blood; viral RNA is only rarely detected in blood [240]. SARS-CoV infected subpleural and peripheral alveoli and a type II cell is the site for its propagation, where the new progeny of viral particles is released, and the cells undergo apoptosis [241]. SARS-CoV- 2 and COVID-19 infection resulted in diffuse alveolar damage (DAD) with fibrin-rich hyaline and releasing of viral particles in free form or vacuolization with double-membrane vesicles that targeting the neighboring new cells/tissues and circulating to reach distant tissues [242]. The virus can be spread reaching many organs and tissues, including the vasculature system [243,244]. Many cellular organelles bound with a lipid membrane that supports their structure and separates them from the cytosol also have an important role in different biochemical processes. To date, all reported positive-strand RNA viruses are characterized by promoting the formation of membranous vesicles, supporting their genomic replication in the cytoplasm [245,246]. In addition, the coronaviruses family stimulates double-membrane vesicles (DMVs) formation with an average diameter of 300 nm [245,246]. These DMVs usually bud from the ER-Golgi intermediate compartment or the endoplasmic reticulum (ER) or ER-Golgi intermediate compartment and enclose the viral particles, newly replicating viral genome, and viral replicase proteins. DMVs diversity may interpret the elapsed cases of reinfected (for your knowledge, we are the first to propose this idea) patients as some of them are asymptomatic with PCR positive and the other exhibited acute to mild COVID-19 symptoms [236,247]. The involvement of such membranous vesicles considers one of the viral strategies to readjust and redirect host cell membranes to participate in viral replication and transcription processes [246,248,249,250]. However, we have discussed the role of exosomes in viral pathogenesis and infection and how several viruses recruit them for their benefit; it should be taken into consideration that exosomes excretion is a normal physiological process and also exhibits a crucial role in protecting host cell against pathogens [11,251]. Recently, it has been reported that isolated exosomes from healthy persons’ semen could inhibit HIV-1 replication in vitro (Madison and Okeoma 2015b) without any effect on HSV-2 or HSV-1 replication [39]. Unfortunately, certain viruses can control these defense mechanisms and recruit them to their profit. 

## 7. Exosomes as a Potent Therapy in Viral Infections

As discussed above, exosomal vesicles constitute a tool of communication between different pathogens and their hosts. Recently, exosomes are considered an attractive area to develop a new drug and can be used as vaccines, drug delivery vehicles, and a diagnostic marker (Figure 3). The possibility to be used as a biomarker for different diseases is attributed to their cargo that differs in diseased and normal conditions and has been identified in all body fluids. Intensive studies are required to explore potential applications of exosomes as diagnostic and prognostic tools in viral infection. Targeted delivery is an area that received great attention as exosomes can be loaded with therapeutic drugs and carry them to specific tissues or organs. In addition, genetically engineered exosomes that are modified to express receptor-specific ligand molecules on their surface are considered a novel system to carry a drug/ miRNA/siRNA-based therapeutic molecule to specific organs. In addition, the incorporation of virus-encoded envelope proteins within genetic material or therapeutic biomolecule into exosomes displaying their superior binding and entry specificity and enhancing their delivery is new [252]. Several studies reported that exosomes can be engineered to express a 29-mer peptide that is generated from the rabies virus glycoprotein (RVG) and specifically binds to acetylcholine receptors on the neurological cell surface. Alvarez-Erviti and coworkers were the first to exploit the RNA-transporting capacity of modified exosomes for transporting small interfering RNAs (siRNA) across the blood–brain barrier. Their study aimed to develop nonimmunogenic targeted delivery of the potential RNA drugs. Targeting was done by transfecting immature dendritic cells (DCs) with plasmids encoding exosomal protein lysosome-associated membrane glycoprotein 2b (Lamp2b), fused with the brain-specific RVG peptide. Purified exosomes were loaded with GAPDH or BACE-1 siRNA and intravenously injected in wild-type mice through the tail. The targeted exosomes specifically delivered GAPDH or BACE-1 siRNA to the mouse brain resulted in a specific knockdown of GAPDH and BACE-1 genes [253]. In addition, engineered exosomes purified from DCs that transduced with adenoviral vector to express FasL, Interleukin (IL)-10, or IL-4 were applied in inflammatory diseases and autoimmune disorders’ treatment [254]. Exosomes are evaluated as cell-free vaccines in infectious diseases. Aline et al. (2004) demonstrated the efficacy of exosomes shed from the DC2.4 cell line as a novel cell-free vaccine inducing protective immunity against toxoplasmosis [255]. Their investigation revealed that exosomes derived from a DC2.4 cell line that had been pulsed ex vivo with *Toxoplasma gondii* are transferred rapidly and preferentially from the intestine, cervical lymph nodes to the spleen and triggered systemic Th1-modulated Toxoplasma-specific immune response in vivo [255]. In addition to their ability to develop protective immunity against Toxoplasma infection, they can be applied as immunoprophylaxis against several viral infections. Exosomes received increasing attention as potential therapeutic modalities (Figure 3)**;** due to their biocompatibility, they offer many advantages, including a means of communication between cells, ability to interact with antigen-presenting cells, representing a good and stable environment for proteins and nucleic acids by protecting them from proteinases, DNase, and RNase, and offering a better distribution tool as they are detected in all bodily fluids [252,256].

It has been reported that exosomes driven from a human monocytic cell line (THP-1) that had been stimulated with lipopolysaccharide endotoxin (LPS) can trigger cytokines such as interleukin 1 beta (IL-1β), tumor necrosis factor-alpha (TNF-α), ligand 5 (CCL5), and chemokine (C-C motif), modulating inflammatory response in healthy mice. In addition, these exosomes were exploited as hepatitis B recombinant antigen adjuvants, as they hastened the appearance of IgG antibody production and enhanced IFN-γ concentration [257]. While these results displayed the capability of the unmodified exosomes to be used as coadjuvants triggering immunostimulatory effect, engineered exosomes can be used as targeted drug delivery, transporting specific molecules for treating different diseases in vivo. Ferrantelli et al. (2018) reported that the product of the fusion between viral antigen and exosome anchoring protein Nef mutant (Nef ^mut^) can be expressed in DNA vector and the strong ability of Nef ^mut^ to accumulate in multivesicular bodies leads to immunogenic exosomes production. The translation of Nef ^mut^–based DNA vector in animal results in engineered exosomes which enclose a huge amount of viral antigen and induce cytotoxic T lymphocyte (CTL) immunization against viral infection. In addition, Ferrantelli et al. [258] reported that the product of the fusion between HBV core protein and exosome anchoring protein Nef ^mut^ can express in DNA vector generated exosomes which encloses huge amounts of HBV core protein and induces cytotoxic T lymphocyte (CTL) immunization against HBV. The advantage of exosomal delivery of specific antiviral molecules makes them a potential therapeutic strategy and/or non-promising vaccine for HBV [258].

In the field of HIV-1, the detection of HIV-1 RNAs and proteins in exosomes derived from the blood of seropositive patients highlights the potential value of exosomes and their cargo as biomarkers for HIV-1. The packaging of HIV-1 components within exosomes derived from the blood of HIV-1-seropositive patients suggests that the body gets rid of viral elements by secreting them into exosomes. The potential of these exosomes as prognostic and diagnostic HIV-1 biomarkers by inducing the shedding of exosomes enclosing the HIV-1 genome to clear the body of viral element and efficient degradation of viral genome can be achieved by engineered exosomes. Recent studies reported that human semen [39,123] and human breast milk-derived exosomes [259] control HIV-1 infection in cell culture thus biofluid exosomes or biofluid exosome-based synthetic nanoparticle delivery systems that enclosing anti-HIV-1 effectors can be applied as potent therapy for AIDS. Moreover, the replication of the murine acquired immunodeficiency syndrome (mAIDS) virus inhibited in the mouse vagina when exposed to human semen-derived exosomes simultaneously with the virus [123]; similar inhibition can occur during sexual transmission when human vaginal mucosa is exposed to HIV-1-seropositive semen. In addition, mice that received the exosome/virus complex showed a reduction in plasma viremia and decreased systemic virus dissemination. These results mark the potential therapeutic effect of semen-derived exosomes on systemic virus spreading and mucosal HIV-1 transmission [123]. The protective properties of milk and semen-derived exosomes against horizontal (sexual) and vertical (milk-borne) HIV-1 transmission made them promising therapeutics for AIDS.

Although the number of clinical investigations to find therapeutic solutions for COVID-19 is growing, no specific vaccine or antiviral treatment is available. Some studies reported that mesenchymal stem cells (MSCs) and/or EVs shed from them can be applied to treat COVID-19 patients owing to their immunomodulatory effect. MSCs or EVs derived from secret different types of cytokines and paracrine factors that suppress the immune system over activation through directly interact with immune cells, including natural killer cells, macrophages, B cells, T cells, and dendritic cells. Furthermore, they secrete and attract different growth factors such as VEGF, TGF, and EGF which improve the microenvironment and promote regeneration of tissue [260]. However, the disadvantages of using MSCs have been reported; for example, the intravenous administration may result in embolus formation, and MSCs derived from embryonic tissue may increase the risk of tumorigenicity and mutagenicity, so the products of MSCs like exosomes [261] and secretome [262] have been recommended as an alternative therapy. Previous studies recommend EVs derived from the cell secretome in the treatment of COVID-19 [263]. In addition, EVs isolated from MSC were reported to produce promising effects as their parental cells with advantages of producing similar or better results than MSCs in vivo investigation, and they are still functional after storing for long periods [261]. By the end of 2020, seven registered clinical trials were using EVs as a strategy to overcome COVID-19 infection [264], and Table 2 displays current clinical trials that apply exosomes in the treatment of COVID-19 infection. Clinical trial NCT04276987 was built to estimate the safety and efficiency of aerosol inhalation of the exosomes isolated from allogeneic adipose MSCs in the treatment of hospitalized COVID-19 patients suffering from coronavirus pneumonia [265] with a protocol assigned to give participated patients conventional treatment and one dose of aerosol inhalation of MSC-derived exosomes (2 × 10^8^ nanovesicles/3 mL) for five uninterrupted days. In the treatment of COVID-19 pneumonia, another clinical trial (ChiCTR2000030261) suggested direct delivery of MSC-derived exosomes into the lungs through the atomization process and the reduction of inflammatory factors and immunity response will be assessed [266]. The clinical trial ChiCTR2000030484 suggested the use of human umbilical mesenchymal stem cells (HUMSCs) and their derived exosomes for the treatment of lung damage in COVID-19 patients [266]. Furthermore, one published study aimed to explore the safety and efficacy of bone marrow MSC-derived exosomes in the treatment of COVID-19 infected patients [267]. After four days of treatment, the laboratory tests and clinical symptoms for these patients were improved without observed adverse effects. The increase in lymphocyte count, the decrease in acute phase markers such as ferritin and C-reactive protein, and the normalization of the neutrophil count were reported.

## 8. Conclusions and Future Perspectives

Data in the present study represent a comprehensive overview of the role of the extracellular vesicles subtype, exosomes in the viral infection, and their pathogenesis mechanisms. Exosomes are a group of nano-extracellular double vesicles formed in the endosomal part of most eukaryotes. Exosomes serve as crucial regulators and mediators of the cell’s communication via transferring their cargo to other cells, thus they have numerous biological purposes. The structure of exosomes is reliant on the tissue and cells’ origin; consequently, they can be different in composition under varied pathological circumstances. Exosomes can be potentially hijacked by many viruses, thus making them influence many processes which are useful for use in many purposes. As a result, viruses utilize biogenesis systems of the hijacked exosome to activate their capsids packaging, regulate their production of virions, and the viral particles secretion. Furthermore, exosomes are being used as nano-carriers of the viral proteins and/or exogenous viral miRNA to be transferred to the non-infected cells. All of these make exosomes have potential importance in several biological functions such as cellular communication and immune modulation, as well as vehicles, to transfer many ingredients from one cell to another. In addition, SARS-CoV-2 is also using exosomes similar to other viruses to be transported for intra-host spreading and viral reproduction. Therefore, exosomes can be suitable candidates for the preparation and development of many viral vaccines for use in the treatment and prevention of many pandemic infections such as COVID-19.

## Figures and Tables

**Figure 1 pharmaceutics-13-01405-f001:**
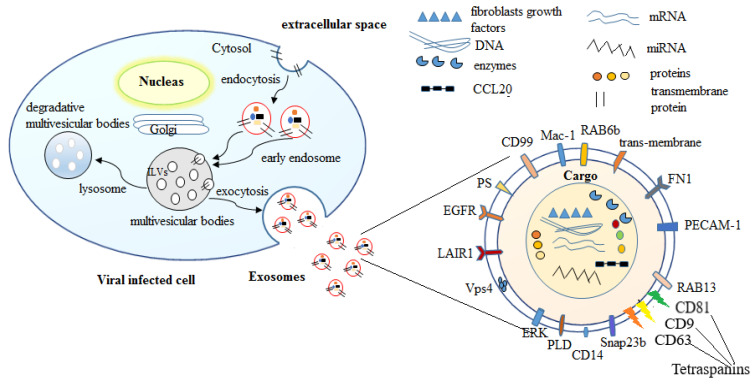
Schematic representation of formation and molecular structure of exosomes. Exosomes are formed by the trafficking of endocytosed proteins to originate the early endosomes through invagination of the plasma membrane. Early endosomes invaginate to generate intraluminal vesicles (ILVs) that are stored inside the multivesicular bodies. These bodies can then fuse with the plasma membrane to generate exosomes which are released into the extracellular space. These exosomes contain proteins, enzymes, nucleic acids (DNA mRNAs, and miRNAs), and lipids. This cargo is surrounded by lipid bilayer membranes which contain some components such as EGFR (epidermal growth factor receptor), LAIR1 (leukocyte-associated immunoglobulin-like receptor 1), Vps4 (Vacuolar protein sorting-associated protein 4), ERK (extracellular-signal-regulated kinase), PLD (phospholipase D), CD14 (cluster of differentiation 14), Snap23 (synaptosomal-associated protein 23b), RAB13 (Ras-related protein Rab-13), PECAM1 (platelet/endothelial cell adhesion molecule 1), FN1 (fibronectin 1), RAB6b (Ras-related protein Rab-6B), Mac-1 (macrophage-1 antigen), CD99 (cluster of differentiation 99), and PS (phosphatidylserine).

**Figure 2 pharmaceutics-13-01405-f002:**
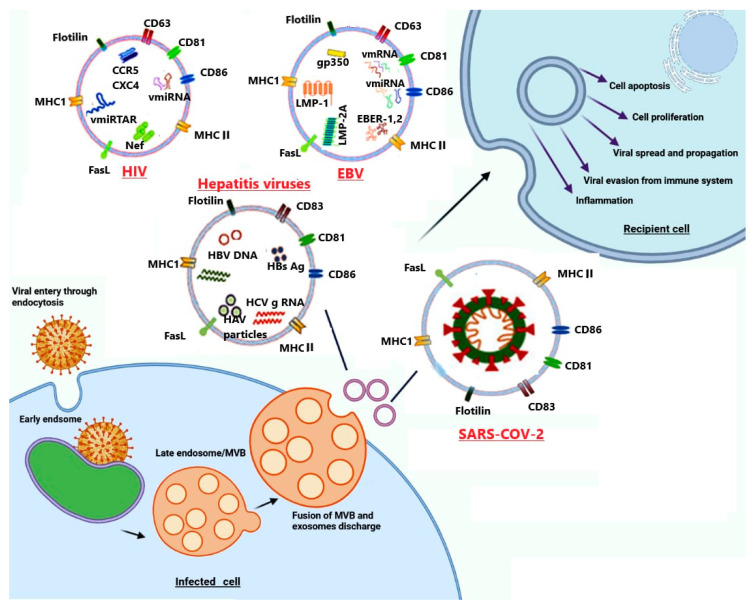
Schematic representation displayed many different viruses hijacking the cellular endocytosis and recruiting exosome as potential mediator in their infection, propagation, and pathogenesis. Here four examples are displayed; HIV, hepatitis viruses, and EBV can export their RNAs and proteins within exosomes. In addition, SARS-CoV-2 can recruit endosomal pathway to export intact virus, their genetic material, or proteins within exosomes.

**Figure 3 pharmaceutics-13-01405-f003:**
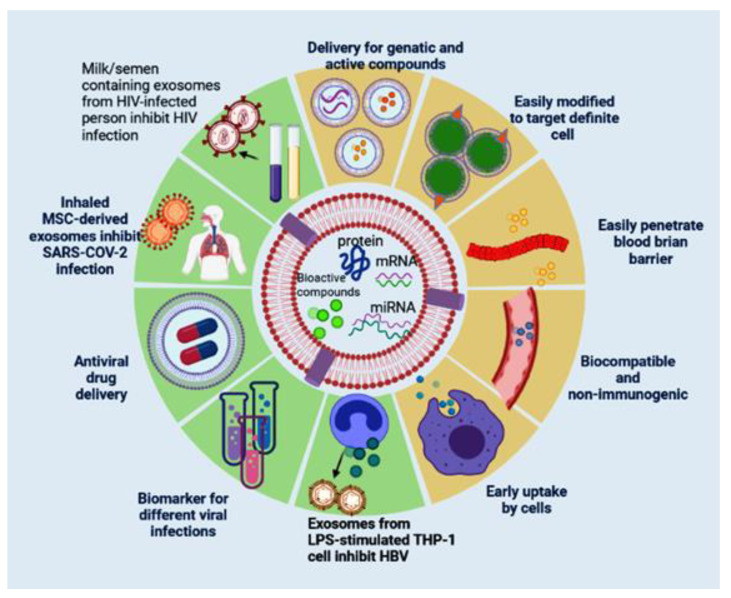
Schematic representation showed the potential exosomal features and their application in different viral treatments. A; display exosomal advantages including delivery for genetic materials, proteins, and drugs, easily modified to target definite cells, biocompatible and non-immunogenic, easily penetrate blood–brain barrier, and easy uptake by cells. B; display their application in different viral treatments such as, HBV, HIV, and SARS-COV-2 (Table 2).

**Table 1 pharmaceutics-13-01405-t001:** Exosomes’ biogenesis and their roles in pathogenesis, medical usefulness, and applications in viral infection.

Viruses	Viral Cargo	Cellular Target	Exosome Biogenesis	Exosomes Roles in the Pathogenesis	Medical Usefulness and Applications	References
Adenovirus	mRNA and miRNA	dendritic cells	Developing of early endosome	Attaching of cell surface receptors onto host cells	The host body of HIV-1 inspires to be clear of viral factors by releasing them into exosomes	[15,77]
HSV-1	VP16, Heat shock proteins, HSV gB, ICP 127, miRNA	Epithelial cells	Trafficking proteins, DNA, RNA and lipids	Delivering of suppressed membrane protein 1 (LMP1) to host cells	Exosomes suppress or stimulate the immune response (immunomodulators)	[15,78,79,80,81,82,83]
Ebola	DNA	macrophage, dendritic cells,	early endosome development	Cell surface receptors Attachment	Clearing the host bodies from virions	[84,85]
EBV	RNA, miRNA, LMP1, 2A, gp350, EBERs	Lymphocytes	budding of endosomal multivesicular bodies	Proliferation, viral reactivation apoptosis, immune evasion	Intercellular communication between cells of the immune system	[86,87,88,89,90,91]
HCV	miRNA, CD9, CD63, CD 81, HCV gRNA, RNA	Hepatocytes	receptor-mediated endocytosis, and plasma membrane fusion	viral maturation and immune evasion	Neutralizing antibodies are resistant to HCV transmission by exosomes as a potential immune evasion mechanism.	[92,93,94]
Dengue virus	Immunoregulator molecules (MHCI and MCII)	Monocytes, macrophages	Recruit ESCRTs to the endosomal membrane	Assembly, transfer of viral RNAs, and suppression of immune activation.	Development of antiviral and vaccine candidate	[95]
HPV	immunoregulator molecules, miRNA	Epithelial cells	ESCRTs are delivered to the site of budding	Apoptosis, viral proliferation,	Promoting anti-apoptotic potential.	[96]
CMV	CMV gB	WBC, epithelial cells	Stimulating membrane budding	Improved viral pathogenicity, infection of myeloid dendritic cells	Inflammatory and regulatory markers	[97]
polyomaviruses, (JCPyV and BKPyV)	Virion particles and miRNA	kidney, bone marrow, and central nervous system (CNS)	Virions packaged within EVs and associated to vesicles surface	play a key role in the dissemination and spread of polyomaviruses	Enhanced viral transmission and can be used as biomarker	[98,99,100,101]
Coxsackievirus B1	Replication competent genome within EV	Epithelial cells	Increased EV biogenesis	Increase viral spread	prolonged viral replication through micro RNA packaged in exosomes and can be used biomarker	[102]
Chikungna virus	apoptotic bodies	Skeletal muscles, brain, and liver cells	hijacking apoptotic bodies	Infectious virions hijacking apoptotic bodies. Increased viral spread	Increased viral spread and can be used as biomarker	[103]
ZIKV	Macrophage-derived exosomes	Brain, skin, placenta, retina, testis, and kidney cells	Infection significantly increases EV formation	Induction of placental proinflammatory cytokine production.	EVs derived from the semen of a ZIKV-infected patient inhibited ZIKV and delivering therapeutics across brain barriers	
	ZIKV NS5-mediated activation of NLRP3			Activation of host inflammatory response and macrophage recruitment promotes inflammation		[104,105]
	EV-bound ZIKV-RNA and E-protein			Increased ZIKV transmission across neurons		[106]
HIV	Cytoskeletal proteins (Actin, Tubulin, Lamin, Myosin)miRTAR, miRNA, Nef	Lymphocytes	ESCRT I and IIenhance membrane budding	Induce proinflammatory cytokines, inhibition of apoptosis, increased susceptibility of naïve T cells, downregulation of CD4 and MHC I, Support viral reproduction and pathogenesis	Analytical indicators in HIV-1, antiviral activity	[107,108,109,110,111]
HBV	HBsAg, DNA, RNA	Hepatocytes	Multivesicular bodies fuse with the plasma membrane and secrete exosomes	Innate immunity evasion, transmission regulators	Drug delivery candidates for the targeted or systematic vehicle to particular organs or tissues	[14,112,113]
HAV	HAV particles, enzymes, HAV gRNA,	Hepatocytes	Transport of ESCRT III, VPS4B, and ALIX. through endosomal-sorting complexes	Increasing viral infectivity, innate immunity evasion, intercellular communications	Drug delivery candidates for the targeted or systematic vehicle to particular organs or tissues	[114]
Rift Valley fever virus	Viral proteins, miRNA, mRNA	WBC	ESCRTs enhance membrane budding	Immune evasion, apoptosis, enhance viral infectivity	Analytic indicators	[115]
human T-lymphotropic virus	mRNA, miRNA, trans-activator protein	Lymphocytes	ESCRTs Stimulate membrane budding	Activate cytokines, damage to neurons, increase viral replication	Contribute to the pathology of the viral infection	[116,117,118]
HHV-8	RNA, miRNA	endothelial cells, WBC	budding of endosomal multivesicular bodies	cell metabolism, immune modulation	Intercellular communication between cells of the immune system	[119,120]

**Table 2 pharmaceutics-13-01405-t002:** Current clinical trials that apply exosomes in the treatment of COVID-19 infection.

Clinical Trials	Applied Therapy	Source	Route of Administration	Outcome/Aims	Ref
ChiCTR2000030484	MSCs and EVs	MSCs derived from the human umbilical cord	Intravenous (IV)	Exploring the safety and efficacy of MSCs and EVs	[266]
ChiCTR2000030261	EVs	MSCs	Aerosol inhalation	Promoting early recovery and avoiding complications through enhancing immunity and inhibiting inflammatory factors	[268]
NCT04389385	EVs	Allogeneic COVID-19 specific T cells (CSTC)	Aerosol inhalation	Estimating the safety and efficiency of inhaled CSTC-exosomes in the treatment of early-stage pneumonia resulting from COVID-19 infection.	[269]
NCT04276987	EVs	MSCs derived from allogeneic adipose tissue	Aerosol inhalation	Exploring the safety and efficiency of inhaled EVs in the treatment of COVID-19 infection.	[270]
NCT04493242	EVs (ExofloTM)	MSCs derived from allogeneic bone marrow	IV	Exploring the safety and efficacy of EVs administrating intravenously as a treatment for ARDS	[271]
NCT04338347	EVs (CAP-1002)	Allogeneic cardiosphere derived cells	IV	Evaluating the safety and efficacy of EVs shed from allogeneic cardiosphere derived cells in the treatment of COVID-19 infection.	[272,273]
NCT04491240	EVs	MSCs	Aerosol inhalation	Estimating the safety and efficiency of exosome inhalation in the treatment of COVID-19 pneumonia	[274]
NCT04384445	HAF, containing EVs (OrganicellTM Flow)	Human amniotic fluid (HAF)	IV	Exploring the safety of HAF-derived acellular products and their efficacy as a therapeutic agent against COVID-19 infection.	[272]

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
