# Peer review of "A Comprehensive Insight into the Role of Exosomes in Viral Infection: Dual Faces Bearing Different Functions"

_pharmaceutics, 2021, doi:10.3390/pharmaceutics13091405_

Round 1
Reviewer 1 Report
Mabroka H. Saad and colleagues reported a comprehensive review on the role of exosomes in viral infection. The issue is foremost important for the comprehension of the relationship of extracellular vesicles machinery and virus features to persist in the host. However, there are some points that should be approached to a more in deep explanation of this matter.
Main points
1- The main point to be approached is the difference among the extracellular vesicles and their features in the purification and extraction from biological fluids and thus their role in viral infection. Numerous studies have reported the role of exosomes in virus infection but the proof of their complete separation from the other extracellular vesicles is extremely hard. Thus, several studies describing viral interaction with exosomes reported are reported as viral interaction with small extracellular vesicles. This is a feature suggested by the “ISEV”. Thus the authors should mention such issue specifying the difference and the discussed problem in the Extracellular vesicles investigation. First, it could be better in the introduction (page 1, line 3) to describe the size of the three type of Extracellular vesicles (exosomes, microvesicles and apoptotic bodies) to clear to the reader the problem.
2- When the point 1 is cleared, the authors should take in consideration to mentioned other virus associated to the use of exosomes or/and small extracellular vesicles (such polyomavirus), at least adding these in table 1.
3- page 4, “exosomes biogenesis and their role …”: in this paragraph it should be described more in deep the mechanism of exosomes generation (role of ESCRT, early endosome, late endosome multivesicular bodies MVB).
4- page 15, lines 5-6: “exosomes can be easily purified..” This is not completely true. In this section it should be added the problems in the separation of the exosomes from the other extracellular vesicles and the methodology used to study their characterization. This is important for the reader to understand the issue.
Minor points
1- Abstract (4 line): Before “However” a full stop should be added.
2- page 2, line 2: Before “Moreover” full stop should be added
3- page 3, figure 1: among proteins associated with exosomes tetraspanin should be added.
4- references associated with studies in table 1 should be checked (for example reference of HAV 45, 167, 228, 229)
5- page 8, line 20: the references should be added with their number.
6- All references list should be checked for the appropriate format.
7 All typos should be corrected.
Author Response
Mabroka H. Saad and colleagues reported a comprehensive review on the role of exosomes in viral infection. The issue is foremost important for the comprehension of the relationship of extracellular vesicles machinery and virus features to persist in the host. However, there are some points that should be approached to a more in deep explanation of this matter.
Main points
- The main point to be approached is the difference among the extracellular vesicles and their features in the purification and extraction from biological fluids and thus their role in viral infection. Numerous studies have reported the role of exosomes in virus infection but the proof of their complete separation from the other extracellular vesicles is extremely hard. Thus, several studies describing viral interaction with exosomes reported are reported as viral interaction with small extracellular vesicles. This is a feature suggested by the “ISEV”. Thus the authors should mention such issue specifying the difference and the discussed problem in the Extracellular vesicles investigation. First, it could be better in the introduction (page 1, line 3) to describe the size of the three type of Extracellular vesicles (exosomes, microvesicles and apoptotic bodies) to clear to the reader the problem.
Thank you for your comments.
We revised these points.
- When the point 1 is cleared, the authors should take in consideration to mentioned other virus associated to the use of exosomes or/and small extracellular vesicles (such polyomavirus), at least adding these in table 1.
Table 1 was revised.
- page 4, “exosomes biogenesis and their role …”: in this paragraph it should be described more in deep the mechanism of exosomes generation (role of ESCRT, early endosome, late endosome multivesicular bodies MVB).
Thank you for your valuable comment.
We revised this section.
- page 15, lines 5-6: “exosomes can be easily purified..” This is not completely true. In this section it should be added the problems in the separation of the exosomes from the other extracellular vesicles and the methodology used to study their characterization. This is important for the reader to understand the issue.
We added section to discuss the purification and characterization of exosomes.
Minor points
- Abstract (4 line): Before “However” a full stop should be added.
We correct this error.
- page 2, line 2: Before “Moreover” full stop should be added
We correct this error.
- page 3, figure 1: among proteins associated with exosomes tetraspanin should be added.
We revised figure 1.
- references associated with studies in table 1 should be checked (for example reference of HAV 45, 167, 228, 229)
We revised table 1.
- page 8, line 20: the references should be added with their number.
We correct this mistake.
- All references list should be checked for the appropriate format.
We revised the references style.
- All typos should be corrected.
We revised the manuscript.
Reviewer 2 Report
The authors provided a well-documented overview of Ev's role in viral infections events. In addition to this, the review could represent a really interesting point of view in a field so dynamic and rich in potential future applications. The field of research focused on exosomes is in continuous evolution and even if the article is well written, the references could be updated with more recent works related to the importance of exosomes in other not viral infections (PMID: 31936232 and similars).
I hope that my comments could be useful and I look forward to reading the revised version of the paper.
Good luck.
Author Response
The authors provided a well-documented overview of Ev's role in viral infections events. In addition to this, the review could represent a really interesting point of view in a field so dynamic and rich in potential future applications. The field of research focused on exosomes is in continuous evolution and even if the article is well written, the references could be updated with more recent works related to the importance of exosomes in other not viral infections (PMID: 31936232 and similars).
Thank you for your comment.
In the present manuscript we focused in the importance of exosomes in viral infection and we in future will write other manuscripts rotated to the role of exosomes in other diseases.
I hope that my comments could be useful and I look forward to reading the revised version of the paper.
The manuscript was revised.
Round 2
Reviewer 1 Report
The Authors have substantially improved the text of the manuscript.
Before the publication the authors should correct few additional points:
1- Page 2: The phrase “Exosomes are extracellular secretory nanovesicles with an estimated density between 1.13 and 1.19 g/ml and their size ranged from 30 to 100 nm [34]” starting the second paragraph should be erased because is unnecessary in this point (it is present at the end of the first paragraph above).
2- Table 1: the reference associated with JC polyomavirus (JCPyV) does not completely describe the use of EVs of such virus (as a vehicle of viral microRNA and virion genome). Additionally, there are also other studies describing EVs carrying microRNA of Polyomavirus or EVs carrying BKPyV. The authors should cite additional other studies for polyomavirus association to the EVs (e.g. Handala et al, J Virol. 2020; Kim MH et al., PLoS One. 2017; Giannecchini S. Viruses. 2020)
3- Page 18, line 32: typos mistake “In” should be correct with “in”.
Author Response
The Authors have substantially improved the text of the manuscript.
Before the publication the authors should correct few additional points:
- Page 2: The phrase “Exosomes are extracellular secretory nanovesicles with an estimated density between 1.13 and 1.19 g/ml and their size ranged from 30 to 100 nm [34]” starting the second paragraph should be erased because is unnecessary in this point (it is present at the end of the first paragraph above).
Thank you for your comment.
We revised this paragraph.
- Table 1: the reference associated with JC polyomavirus (JCPyV) does not completely describe the use of EVs of such virus (as a vehicle of viral microRNA and virion genome). Additionally, there are also other studies describing EVs carrying microRNA of Polyomavirus or EVs carrying BKPyV. The authors should cite additional other studies for polyomavirus association to the EVs (e.g. Handala et al, J Virol. 2020; Kim MH et al., PLoS One. 2017; Giannecchini S. Viruses. 2020)
3- Page 18, line 32: typos mistake “In” should be correct with “in”.
Thank you for your comment.
We added these references.
